# Tdap Booster Vaccination for Adults: Real-World Adherence to Current Recommendations in Italy and Evaluation of Two Alternative Strategies

**DOI:** 10.3390/ijerph19074066

**Published:** 2022-03-29

**Authors:** Maurizio Lecce, Pier Mario Perrone, Silvana Castaldi

**Affiliations:** 1Department Biomedical Sciences for Health, Postgraduate School in Public Health, University of Milan, 20136 Milan, Italy; maurizio.lecce@unimi.it (M.L.); silvana.castaldi@unimi.it (S.C.); 2Quality Unit Fondazione IRCCS Ca’ Granda OMP, 20122 Milan, Italy

**Keywords:** tetanus-diphtheria-pertussis vaccination, adult Tdap booster, healthcare services’ adherence

## Abstract

Background: While the effectiveness of tetanus-diphtheria-pertussis childhood immunization programs is unquestionable, the actual need for a periodic boosting vaccination in adults is controversial. In Italy, the Ministry of Health recommends a Tdap booster vaccination every 10 years. The aim of this study is to assess the real-world adherence of Italian regional healthcare services to national recommendations and to evaluate two alternative strategies. Methods: Annual Tdap vaccine requirements by the 21 Italian regions were retrieved from related tender announcements, and regional and national vaccination coverage rates (VCRs) were estimated for three scenarios, namely the currently recommended 10-year booster vaccination, a single booster shot at age 50 and at age 65. Results: In Scenario 1, no region reached a VCR > 30%, and the national VCR was 10.6%; in Scenario 2, five regions achieved the optimal vaccination coverage of ≥95%, but the vast majority continued to have inadequate VCRs, with a national VCR of 54.4%; in Scenario 3, five regions reached VCRs exceeding 100%, with VCRs from other regions significantly improving and a national VCR of 74.8%. Conclusions: A substantial lack of adherence by Italian regional healthcare services to current national recommendations on tetanus-diphtheria-pertussis adult vaccination was shown. Scenario 3 is the most feasible, i.e., a single booster shot at age 65, possibly administrable along with other already-recommended, age-specific vaccines.

## 1. Introduction

Tetanus (T), diphtheria (D) and pertussis (P) are three vaccine-preventable diseases, and the implementation of large-scale childhood immunization programs with TDP vaccine since the 1950s has resulted in a dramatic decline in incidence and mortality rates of all three diseases [1,2,3].

As a direct consequence, the epidemiology of TDP progressively changed, with a shift in the age distribution from infants and children toward older age groups, i.e., adolescents, adults and elderly, especially in developed countries [4,5,6]. Concerning Europe, the latest (2018) ECDC epidemiological report on T showed that individuals aged ≥ 65 years were the most affected, accounting for 75% of all reported cases [7]. An analog report on D showed that the largest proportion of cases due to *Corynebacterium ulcerans*, which are more likely to be indigenous, occurred in individuals aged ≥ 65 years, while cases due to *C. diphtheriae* occurred in younger age groups, but 60% of them were imported from developing countries with disrupted immunization programs [8]. Lastly, the 2018 ECDC report on P showed that the largest number of cases occurred in adolescents and adults, representing the main source of infection for infants < 1 year of age, who are still the most affected age group in terms of morbidity and mortality [9].

Epidemiological data are confirmed and integrated by a wealth of seroprevalence studies revealing that the percentage of individuals without protective antibody levels against T and D increases with age, and it is highest for older adults and elderly [10,11,12]. Serosurveys also demonstrate the persistent circulation of *Bordetella pertussis* in adolescents and adults in Europe, with a negligible proportion of individuals not having acquired the infection recently [13,14].

These findings could be explained by several factors; however, the waning of the vaccine-induced antibody response against TDP is well recognized and largely accepted by the scientific community [4,5,6,15,16,17].

As a result, most countries in North America and Europe have issued recommendations for adults (i.e., ≥19 years of age) to receive periodic booster doses of tetanus toxoid-reduced diphtheria toxoid-acellular pertussis (Tdap) vaccine with the aim to maintain high immunity levels against TDP in all age groups [15,18,19].

However, booster vaccination policy for adults remains controversial [20]: the main arguments against this approach are the absence of official recommendations for this age group by WHO [4,5,6], low incidence rates of T and D in North America [21,22,23] and Europe [7,8,9,24], likely greater effectiveness of maternal immunization in preventing infant P rather than generalized adults vaccination or the cocooning strategy [6,25], and sustained durability of the vaccine-induced antibody response against T and/or D with apparent protection through most of life [14,26,27,28]. Indeed, some authors proposed to abandon Tdap periodic booster vaccination for adults [29] and/or to consider a single midlife booster shot at age 50–65 years [20,26,30], as already implemented in some European countries, e.g., Spain, Poland and Croatia [19].

Italy recommends Tdap booster vaccination every 10 years for all adults aged ≥19 years according to the 2017–2019 National Immunization Prevention Plan [31] and the related National Immunization Schedule [32], whose validity was postponed throughout 2021 due to the current COVID-19 pandemic [33].

However, whereas DTaP vaccination coverage rates in children and adolescents are routinely recorded, no systematic documentation of Tdap uptake among adults is in place. Tdap coverage data in adults are similarly scarce or inconsistent all across Europe [34]. A seroprevalence study performed in 2015 in 6 European countries reported for Italy a Tdap vaccination coverage among adults of only 38% [11].

The aim of this study is to evaluate the general “state of the art” of Tdap booster vaccination among adults in Italy, testing the comprehensive adherence of the system to the current recommendations and further exploring two alternative scenarios, namely a single booster shot at age 50 and at age 65.

## 2. Materials and Methods

We analyzed tender announcements issued by Italian regions concerning the Tdap vaccine for adults and retrieved the annual amounts required. We focused on requirements referring to 2019, i.e., the pre-pandemic period, to avoid possible data distortions due to the disruption of routine vaccination services caused by the pandemic. For the Abruzzo and Molise regions, a unique tender announcement was issued, and only aggregated data were available for Tdap, but we estimated vaccine requirements from each region considering T toxoid—reduced D toxoid (Td) vaccine requirements in the same tender. For Valle d’Aosta and the Autonomous Province of Bolzano (P.A. Bolzano), no data were available for 2019, and we used Tdap requirements for 2020. Similarly, for Umbria, no data for 2019 could be retrieved, and we estimated the 2019 Tdap requirement by mean between requirements from the previous (2015) and the following (2020) tender. Population data on 1 January, 2019, from the Italian Institute of Statistics (ISTAT) database gave the three cohorts of interest, i.e., population aged ≥ 19 years, aged 50 years and aged 65 years. We calculated estimated vaccine coverage rates (VCRs) in these three scenarios. For the first scenario, as Tdap booster vaccination is recommended every 10 years, we obtained the number of adults who every year should receive a Tdap booster by dividing the population cohort aged ≥ 19 years by 10. Finally, we calculated the estimated national Tdap vaccination coverage rate by means of regional VCRs for all three scenarios.

## 3. Results

In Table 1 Tdap vaccine requirements by the 21 Italian regions for the year 2019 are shown with the five exemptions mentioned above (Abruzzo, Molise, Valle d’Aosta, P.A Bolzano, Umbria), along with the size of the three cohorts of interest and the respective estimated vaccination coverage rates.

In Scenario 1, no region reached the threshold for optimal vaccine coverage rate, i.e., ≥95%. Eight regions (38.1%), i.e., Piemonte, Liguria, P.A. Bolzano, F.V.G., Toscana, Calabria, Emilia Romagna and Marche, reached a VCR ≥ 10%, and only two reached a VCR > 20%, namely Emilia Romagna and Calabria, with VCRs of 27.8% and 26.9%, respectively. In all the other 13 regions (61.9%), the achieved VCRs were extremely low, i.e., <10%. In this scenario, the estimated national VCR was 10.6%.

In Scenario 2, namely a single booster shot for adults aged 50 years, 5 regions (23.8%) achieved optimal vaccination coverage, i.e., Calabria, Emilia Romagna, Friuli Venezia Giulia, Toscana and P.A. Bolzano. Furthermore, 2 regions, i.e., Calabria and Emilia Romagna, had their availability vaccine amounts even overcoming the actual cohort numerosity (VCR of 147.8% and 139.5%, respectively). Twelve regions (57.1%) achieved VCRs between 20% and 70%. The remaining 4 regions (19%) reached VCRs < 20%. In this setting, the estimated national coverage rate was 54.4%.

Scenario 3 refers to a single Tdap booster dose for the 65-year-old cohort. This represents the most favorable setting, as 5 regions (23.8%), i.e., Emilia Romagna, Calabria, P.A. Bolzano, Toscana and Friuli Venezia Giulia, reached a VCR ≥ 100%, or in other words, had vaccine amounts higher than those needed to vaccinate all individuals aged 65 years. The Marche region achieved a VCR of 92%, Piemonte 84%, Liguria 81.9% and Lombardia 71%. Four regions (19%) reached moderate VCRs ranging from 51.1 to 66.3%. The remaining 8 regions (38.1%) continued to achieve insufficient vaccination coverage ranging from 10.9% to 32.1%. In Scenario 3, the highest national Tdap coverage rate was achieved, i.e., 74.8%.

## 4. Discussion

This study demonstrates a substantial lack of adherence by the regional health services in Italy to the current National Immunization Prevention Plan’s recommendations for immunization against TDP in the adult population, i.e., a Tdap booster shot every 10 years for all adults aged ≥19 years. In this setting, all the 21 Italian regions show inadequate estimated vaccination coverage rates, with only two of them achieving a VCR > 20%, and 13 regions reaching completely insufficient VCRs, i.e., <10%. It is important to notice that no geographical trends seem to be present, as of the two regions with the highest coverage rates, one is placed in Northern Italy and one in Southern Italy. Moreover, of the 6 regions with VCRs between 10 and 20%, some are placed in Northern and some in Central Italy. Concerning the 13 regions with VCRs < 10%, they are indistinctly located in Northern, Central and Southern Italy. Reducing the size of the eligible population, namely considering a single Tdap booster vaccination for the population cohort aged 50 years, VCRs expectedly increased, and 3 regions reached the optimal coverage rate of ≥95%, namely Friuli Venezia Giulia, P.A. Bolzano and Toscana. Furthermore, the two already mentioned regions Emilia Romagna and Calabria had vaccine amounts exceeding those needed. The three regions Marche, Piemonte and Liguria reached moderate coverage rates ranging from 60% to 70%, while the other 13 regions continued to have inadequate coverage rates, i.e., <50%. The best scenario is the third one, corresponding to a single Tdap booster shot for adults aged 65 years. In this setting, 9 regions reached moderate to high or very high coverage rates, i.e., Lombardia (71%), Liguria (81.9%), Piemonte (84%), Marche (92%), and 5 of them, i.e., Emilia Romagna, Calabria, P.A. Bolzano, Toscana and Friuli Venezia Giulia, could even reduce their annual requirements due to exceeding stocks. In this setting, a more consistent North–South geographical gradient could be noticed, as of these 9 regions, 6 (66.6%) are located in Northern Italy, 2 (33.3%) in Central Italy and 1 (16.6%) in Southern Italy. Otherwise, it could be noticed that even in the most favorable setting, 12 regions (57.1%) continued to show moderate to low or very low coverage rates ranging from 66.3% of Lazio to 10.9% of Molise, demonstrating insufficient Tdap vaccine requirements.

The decision on whether to recommend a decennial Tdap booster vaccination rather than switching to a single booster shot in older adults is up to the Italian Ministry of Health, with several reasons supporting either the first or the second vaccination approach.

In Italy, the current epidemiology for T and D is characterized by low incidence rates. According to ECDC, a mean of 39.2 T cases per year have been notified in the time period 2014–2018 (max. = 49 in 2014, min. = 30 in 2016), with an age-standardized incidence rate of 0.04/100.000 in 2018 [7]. Interestingly, of the 478 T cases that occurred in 2014–2018, in 26 EU/EEA countries, Italy accounted for 196 (41%) of them [7]. However, it should be clarified that Italy uses a clinical case definition for notification, while in most European countries, notified cases need to be laboratory-confirmed [7]. A recent Italian study investigating T epidemiology in 2001–2010, although having ascertained quite under-reporting of cases by healthcare providers, estimated equally low incidence rates, with a mean of 93.2 T cases per year corresponding to an average incidence rate of 1.6/1.000.000/year and 21 deaths per year [10]. Interestingly, both this Italian study and ECDC found that the highest proportion of cases occurred in individuals aged 65 years and over, namely 80% and 91.7%, respectively.

Concerning D, ECDC reported one notified case in Italy in 2018, one in 2017, one in 2016, no cases in 2015 and one in 2014 [8]. Likewise, incidence rates seem consistently very low all across Europe, as 63 cases were overall notified in 2018 [8]. Of them, 33 cases were due to *C. ulcerans*, which is more likely to be an indigenous pathogen, and the largest proportion occurred in people aged ≥ 65 years; 29 cases were due to *C. diphtheriae* and occurred in younger age groups; however, 60% of them were imported cases of travelers or migrants from countries where the infection is endemic [8]. These epidemiological data and trends are further confirmed by other authors [24,35,36].

Together, these findings seem to suggest that, at least for T and D vaccination, a booster vaccination every 10 years might no longer be necessary and that a single booster vaccination at age 50–65 might be sufficient.

On the other hand, several serologic studies found a wider susceptibility to T and even more to D across the population including younger age groups. A national population-based seroprevalence study conducted in Italy in 2003–2004 showed that only 77.8% of individuals aged 25–44 years and 43.4% of individuals aged 45–64 years had protective antibody titers against T (i.e., >0.1 IU/mL) [10]. A serosurvey in 2016–2017 among 509 healthcare workers in Catalonia, Spain, showed sufficient protection against T in younger age groups but percentages of protected individuals against D of only 66.7% and 55.4% in the age groups of 35–44 years and 45–54 years, respectively [12]. Grasse et al. in 2010 compared immunological response following Tdap immunization in an elderly (range = 66–92 years of age) and a young cohort (range 24–40 years of age), finding that in the latter, 100% were protected against T, but 52% were not protected against D, and that 5 years after immunization, sufficient protection was still present for T, but 24% were again not protected against D [17]. In the 1990s, a large D outbreak occurred in the Russian Federation and former Soviet Republics, with more than 157,000 cases and 5000 deaths reported, suggesting its potential for a persistent circulation and possible resurgence [37,38]. Extensive investigations suggested a combination of several causative factors, including increased susceptibility of both adolescents and adults and the deterioration of national immunization programs [37]. Together, these data suggest that a decennial booster vaccination in all adults aged ≥ 19 years might still be necessary to ensure protective antibody levels against T and particularly against D across all age groups.

Concerning P, the disease burden is much greater compared to T and D, with even more peculiar epidemiology. Effective childhood immunization programs since the 1950s have led to a steep decline in disease incidence among children and to a significant age shift toward adolescents and adults, particularly in high-income countries [6]. However, while these age groups often develop a mild or even asymptomatic infection with consequent underdiagnosis and under-reporting [39], they act as a source of infection for infants <1 year of age who are too young to receive the primary DTaP vaccination, leading to an excess in morbidity and mortality rates in this age group [3,6]. According to ECDC, in 2018, there were 35,627 P cases reported by 30 EU/EEA countries, with an overall notification rate of 8.2/100,000 population [9]. However, individuals aged ≥ 15 years accounted for 62% of all cases reported, and infants aged < 1 year were the most affected age group, with the highest rate of 44.4/100,000 population and the largest proportion of those hospitalized [9]. Concerning Italy, 900 confirmed P cases occurred in 2018 (age-standardized notification rate of 1.8 per 100,000 population), with a substantially stable trend [9]. These epidemiological data are largely confirmed by several serologic studies revealing that *Bordetella pertussis* is still circulating among adolescents and adults all across Europe. In a seroprevalence study performed in 2005 in France among healthy individuals aged 18–60 years attending vaccination centers for travelers, 7.6% of the participants showed anti-P antibody titers ≥125 UI/mL indicative of a recent P infection, and this percentage raised to 13.4% in the 18–29 years age group [13]. A large serosurvey conducted in 2015–2018 in 18 European countries among two adult cohorts aged 40–49 and 50–59 years found a not negligible proportion of the study population ranging from 2.7 to 5.8% showed P-specific antibody levels > 100 UI/mL, thus revealing a recent exposure to *Bordetella pertussis*. Whereas the above-mentioned data would suggest the still need for a periodic Tdap booster vaccination in both young and older adults, the limited effectiveness of this approach in preventing severe P in infants was shown [6]. Likewise, the “cocooning” strategy introduced in the early 2000s in some high-income countries, i.e., protecting infants too young to be vaccinated through vaccination of their close contacts (siblings, parents, grandparents, etc.), only proved to provide a limited benefit [40,41]. On the contrary, maternal immunization with a single Tdap booster in the third trimester of pregnancy with the aim to transfer maternal antibodies to the unprotected newborn proved effective in preventing P in the first weeks/months of life [25], and it is currently recommended by WHO [6].

The aim of our study is not to take a firm stand in favor of decennial Tdap vaccination rather than a single midlife booster vaccination policy, but to provide a concrete report on the real-world adherence to the current recommendations in the Italian context, as well as to perform a “simulation” in case the midlife booster approach would be preferred.

From a methodology viewpoint, it is important to underline that the earliest tender announcements on the Tdap vaccine retrievable from publicly accessible sources of information (in most cases, regional healthcare services’ websites) were indeed issued in 2019 or 2020. Furthermore, they could refer to a single year (e.g., only 2019, only 2020) or to a wider range (e.g., 2019–2022, 2020–2023). Anyway, we decided to focus on Tdap requirements for the year 2019, mainly because it represented the year just before the COVID-19 pandemic explosion, and we wanted to assess the regional healthcare services’ adherence to national recommendations in a “steady-state” period and not in an emergency context without avoidable data distortions due to the unquestionable reduction/disruption of the routine vaccination services. Furthermore, 2019 was also the year for which we have data from the vast majority of regional healthcare services. Finally, even focusing on 2020 tenders, most were issued in the first part of the year, so it is very unlikely that they took into account the imminent changes due to COVID-19; on the contrary, it is likely that requirements were made based on the historical demand. As for all three explored scenarios, the optimal vaccination coverage rate of ≥95% is far from being achieved (for all regions in Scenario 1, and for 16 regions in Scenarios 2 and 3, respectively), and it is important to elaborate on some context-specific strategies in order to significantly increase the vaccination coverage of the eligible population. As for Scenario 1, all adults aged ≥ 19 years should receive a Tdap booster vaccination every 10 years, and a routine 10-year medical examination for driving license renewal could be an excellent opportunity to administer the Tdap booster dose, as well as to check the individual’s Tdap immunization status and eventually proceed to catch-up vaccination. Concerning Scenario 2, the single-dose Tdap booster shot at age 50 could coincide with the active call by healthcare services in the setting of the colorectal and breast cancer screening program, since in Italy, screening for colorectal and breast cancer is offered actively and free of charge to all individuals aged 50–69 years [42,43]. As for Scenario 3, a single-dose Tdap booster shot at age 65 could be administered together with other vaccines already recommended for individuals aged ≥ 65 years by the Italian National Immunization Prevention Plan, namely the influenza vaccine, the herpes zoster vaccine and the pneumococcal vaccine [31,32].

As for Scenario 2 (single booster shot at age 50) and Scenario 3 (single booster shot at age 65), the estimated national Tdap coverage rate progressively improves, and a very interesting question would be if infection rates by T, D and P pathogens would be affected in the event that one of these two approaches is actually adopted. Several factors should be considered: concerning T and D, the current low incidence rates despite poor Tdap vaccination coverage for the Italian adult population (38% according to a recent study [11]) might suggest that valid protection could already be provided by the childhood vaccination cycle. Moreover, the fact that the vast majority of T cases occur in individuals aged 65 and above [7,10] might suggest that a waning immunity plays an important role [4,5,6,15,16,17]; therefore, a single midlife booster shot would be a timely intervention to prolong immune protection. Concerning P, given that infants aged < 1 year are the most affected age group [9] and that the periodic Tdap booster vaccination in both young and older adults, as well as the “cocooning” strategy, proved of limited effectiveness in preventing severe infant P [6,40,41] compared to the maternal immunization [6,25], it might suggest that a single midlife booster shot in place of the decennial booster vaccination would not lead to an increase in P incidence and mortality rates for this age group. Finally, it could also be argued that a single appointment for vaccination (at age 50 or 65), eventually linked to other events as just discussed, could be helpful in “concentrating” vaccination efforts on a circumscribed population at a circumscribed time rather than a “dispersion” throughout adulthood, thus effectively increasing the population’s immunity levels. While all these points seem to suggest that a single midlife booster shot would not negatively affect DTP infection rates at all, the waning immunity phenomenon, together with plenty of serologic studies demonstrating the population’s underprotection and persistent circulation of these pathogens, could be equally valid arguments in favor of a possible increase in DTP infection rates. For all these reasons, further research on the actual Tdap-induced immune protection in terms of efficacy and durability is strongly needed.

Finally, the main limitation in this study is that the Tdap vaccine requirements in regional tender announcements are estimates of the needed amounts of Tdap vaccine. Therefore, they could not correspond to the number of booster doses actually administered for the year 2019. As a consequence, the calculated regional and national VCRs in all three scenarios are equally to be considered as estimated and not punctual. On the other hand, we believe that this could be a valuable method to obtain a good proxy of the actual vaccination coverage and consequent adherence to current recommendations in a given country, far exceeding the validity of other routinely used methods, e.g., self or investigator-administered questionnaires, which is also the main strength of the study.

## 5. Conclusions

This study demonstrates the substantial lack of adherence by the Italian regional health services to the current National Immunization Prevention Plan’s recommendations on Tdap immunization for adults, with estimated regional and overall vaccination coverage rates far from reaching the optimal coverage threshold. Two alternative scenarios have been explored, given the current controversy about the Tdap vaccination schedule for adults [20,26,29,30]. In the hypothesis of a single Tdap booster vaccination at age 50 (Scenario 2), 5 regions achieved the optimal VCR of ≥95%. This alternative vaccination strategy could be linked to the active and free-of-charge screening programs for colorectal and breast cancer [42,43]. In the hypothesis of a single Tdap booster administered at age 65 (Scenario 3), regional VCRs would significantly improve, and a national VCR of 74.8% would be achieved. This latter strategy, which represents the most feasible one, could be implemented together with other elderly-specific vaccine programs [31,32].

## Figures and Tables

**Table 1 ijerph-19-04066-t001:** Tdap requirements from 21 Italian Regions for year 2019, eligible populations, regional and national VCRs.

Regions	Tdap Requirem. 2019	Age 19+	VCR%	Age 50	VCR%	Age 65	VCR%
Piemonte	45,000	364,043	12.4	71,383	63.0	53,562	84.0
Valle d’Aosta	500 **	10,447	4.8	2127	23.5	1575	31.7
Liguria	15,936	131,193	12.1	25,474	62.6	19,464	81.9
Lombardia	80,000	827,378	9.7	167,507	47.8	112,629	71.0
P.A. Bolzano	8340 **	42,390	19.7	8779	95.0	5324	156.6
P.A. Trento	1666	44,358	3.8	8697	19.2	6429	25.9
Veneto	16,865	405,227	4.2	82,514	20.4	57,149	29.5
F.V.G.	19,953	102,282	19.5	19,999	99.8	15,116	132.0
Emilia Romagna	103,333	371,785	27.8	74,060	139.5	50,849	203.2
Toscana	58,180	311,409	18.7	60,301	96.5	43,890	132.6
Umbria	6352 ***	73,441	8.6	13,845	45.9	10,753	59.1
Marche	16,800	127,370	13.2	24,160	69.5	18,254	92.0
Lazio	43,700	479,448	9.1	97,163	45.0	65,903	66.3
Abruzzo	4886 *	109,291	4.5	20,884	23.4	15,790	30.9
Molise	414 *	25,857	1.6	4730	8.8	3807	10.9
Campania	37,200	464,691	8.0	88,801	41.9	62,912	59.1
Puglia	24,000	328,856	7.3	61,880	38.8	46,983	51.1
Basilicata	960	47,021	2.0	8776	10.9	6881	14.0
Calabria	42,500	158,235	26.9	28,757	147.8	23,206	183.1
Sicilia	18,166	402,439	4.5	74,695	24.3	56,576	32.1
Sardegna	5000	138,504	3.6	26,420	18.9	21,635	23.1
Total	549,751	4,965,665	10.6	970,952	54.4	698,687	74.8

* Only aggregated data available for Tdap: requirements per single region were estimated considering Td vaccine requirements in the same tender; ** no data available for 2019: Tdap requirements for 2020 were used; *** no data available for 2019: Tdap requirements were estimated by mean between 2015 and 2020 Tdap tender announcements.

## Data Availability

Not applicable.

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
