# Peer review of "Tdap Booster Vaccination for Adults: Real-World Adherence to Current Recommendations in Italy and Evaluation of Two Alternative Strategies"

_ijerph, 2022, doi:10.3390/ijerph19074066_

Round 1

Reviewer 1 Report

In this study Lecce and colleagues showed that the Italian Regional healthcare services lack of adherence to current national recommendations on tetanus-diphtheria-pertussis (DTP) adult vaccination . Following 3 scenario, scenario n. 1 (aged 19+), scenario n. 2 ( aged 50), and scenario n. 3 (aged 65), the authors showed that the latter strategy (Single dose to age 65) represents the most feasible startegy, and It could be implemented together with other elderly-specific vaccine programmes

In general the manuscript is good, I have some suggestions

1- page 3: results please define ( scenario n. 1)

2- Discussion the authors needs to do the followings

a) Most data from 2019 or 2020. What about other data from Italy from earlier years? Did the authors analyze them? Is the conclusion same?

b) What about other European countries?

c) What is the risk of developing diphteria, tetanus or pertussis in age between 19-65 in Italy ?

if the maunscript is published and decison is taken to give a single booster dose at age 50 or 65, does this decision affect infection rate by7 these pathogens?

Author Response

Dear Reviewer, 

I would like to thank you for your comments and suggestions which guided us through a review process that modified the article. I uploaded the new version of the article and I’ll try to give an explanation to all your comments point by point.

In this study Lecce and colleagues showed that the Italian Regional healthcare services lack of adherence to current national recommendations on tetanus-diphtheria-pertussis (DTP) adult vaccination. Following 3 scenario, scenario n. 1 (aged 19+), scenario n. 2 ( aged 50), and scenario n. 3 (aged 65), the authors showed that the latter strategy (Single dose to age 65) represents the most feasible strategy, and It could be implemented together with other elderly-specific vaccine programmes

In general the manuscript is good, I have some suggestions

1- page 3: results please define ( scenario n. 1)

We have better defined the results underline specifically the various regions with different VCR

2- Discussion the authors needs to do the followings

  1. a) Most data from 2019 or 2020. What about other data from Italy from earlier years? Did the authors analyze them? Is the conclusion same?

Thank you so much for this point. We managed to add a paragraph in the Discussion.

“From a methodology viewpoint, it is important to underline that earliest tender announcements on Tdap vaccine retrievable from publicly-accessible sources of information (in most cases, Regional healthcare services’ websites) were indeed issued in 2019 or 2020. Furthermore, they could refer to a single year (e.g., only 2019, only 2020) or to a wider range (e.g. 2019-2022, 2020-2023). Anyway, we decided to focus on Tdap requirements for the year 2019 mainly because it represented the year just before the COVID-19 pandemic explosion and we wanted to assess the Regional healthcare services’ adherence to national recommendations in a “steady state” period and not in an emergency context with not avoidable data distortions due to the unquestionable reduction/disruption of the routine vaccination services. Furthermore, 2019 was also the year for which we have data from the vast majority of Regional healthcare services. Finally, even focusing on 2020 tenders, most were issued in the first part of the year, so it is very unlikely that they took into account the imminent changes due to COVID-19, on the contrary it is likely that requirements were made based on the historical demand”.

  1. b) What about other European countries?

We think it could be very difficult to retrieve tender announcements on a given vaccine from other European countries, mainly because they are very specific information, likely to be retrievable in local healthcare authorities’ sources of information, likely in native language etc. Neither it was our purpose to focus on another EU country’s vaccine requirements, because we wanted to mainly focus on the Italian healthcare system. Otherwise it is easier to consider Tdap vaccine coverage rates in a given country, and of course we made an extensive research for these data but, as stated in the final part of Introduction, Tdap coverage data in adults are scarce or inconsistent not only in Italy but all across Europe.  

  1. c) What is the risk of developing diphteria, tetanus or pertussis in age between 19-65 in Italy?

We do not have a definitive answer about that. We believe we could give a good proxy of this risk reporting (in Discussion) that: for tetanus, the most recent data (ECDC report in 2018) indicate in Italy an age-standardized notification rate of 0.04 per 100.000 population and that 91.7% cases occur in the age group aged 65 years and above; for diphtheria, that latest ECDC report describes only 1 case notified in 2018 in Italy, therefore it can be assumed an infinitesimal risk; for pertussis, that 900 confirmed pertussis cases occurred in 2018 in Italy – we added the age-standardized notification rate of 1.8 per 100.000 population in the manuscript – and, on a European scale, that 62% of notified cases occurred in people aged 15 years, but individuals from this age groups often develop a mild or even asymptomatic infection with consequent under-diagnosis and under-reporting.

If the manuscript is published and decision is taken to give a single booster dose at age 50 or 65, does this decision affect infection rate by these pathogens?

That is a really good question, thank you. We added the following paragraph in the Discussion:

“As in scenario n. 2 (single booster shot at age 50) and scenario n. 3 (single booster shot at age 65) the estimated national Tdap coverage rate progressively improves, a very interesting question would be if infection rates by T, D and P pathogens would be affected, in the event that one of these two approaches is actually adopted. Several factors should be considered: concerning T and D, the current low incidence rates despite a poor Tdap vaccination coverage for the Italian adult population (38% according to a recent study [11]) might suggest that a valid protection could already be provided by the childhood vaccination cycle. Moreover, the fact that the vast majority of T cases occur in individuals aged 65 and above [7,10] might suggest that a waning immunity plays an important role [4-6, 15-17], therefore a single midlife booster shot would be a timely intervention to prolong immune protection. Concerning P, given that infants aged < 1 year are the most affected age group [9] and that the periodic Tdap booster vaccination in both young and older adults as well as the “cocooning” strategy proved of limited effectiveness in preventing severe infant P [6,40,41] compared to the maternal immunization [6,25], it might suggest that a single midlife booster shot in place of the decennial booster vaccination would not lead to an increase of P incidence and mortality rates for this age group. Finally, it could also be argued that a single appointment for vaccination (at age 50 or 65) – eventually linked to other events as just discussed – could be helpful in “concentrating” vaccination efforts on a circumscribed population at a circumscribed time rather than a “dispersion” throughout adulthood, thus effectively increasing population’s immunity levels. Whilst all these points seem to suggest that a single midlife booster shot would not negatively affect DTP infection rates at all, the waning immunity phenomenon together with plenty of serologic studies demonstrating population’s under-protection and persistent circulation of these pathogens could be equally valid arguments in favor of a possible increase of DTP infection rates. For all these reasons further research on the actual Tdap-induced immune protection in terms of efficacy and durability is strongly needed.”.

Reviewer 2 Report

  • Query 1: Line 15 - the word 'national' is misspelled.
  • Query 2: Lines 119-127. The scenario 3 is described in 20 regions instead of 21. Is there any reason why Lombardia (VCR 71%) was not included?
  • Query 3: Line 131. I suggest rewriting this sentence.
  • Query 4: Line 311. I suggest deleting the last character (Ù).
  • Query 5: I suggest deleting 'Published 2020 Feb 13'.
  • Query 6: I recommend checking the use of commas, as there are some missing commas, especially in the discussion.

Author Response

Dear Reviewer, 

I would like to thank you for your comments and suggestions which guided us through a review process that modified the article. I uploaded the new version of the article and I’ll try to give an explanation to all your comments point by point.

Query 1: Line 15 - the word 'national' is misspelled.

We proceeded to correct the word national adding the missing i at line 15

Query 2: Lines 119-127. The scenario 3 is described in 20 regions instead of 21. Is there any reason why Lombardia (VCR 71%) was not included?

No reasons, it’s a mistake, so we have included Lombardia to the five regions with VCR between 70% and 92%

Query 3: Line 131. I suggest rewriting this sentence.

We proceeded to rewriting the sentence; the previous one was wrong due to  a mispositioning during the review process before submission

Query 4: Line 311. I suggest deleting the last character (Ù).

We proceeded to delete the last character at line 311

Query 5: I suggest deleting 'Published 2020 Feb 13'.

We proceeded to delete ‘Published 2020 feb 13’ at lines 346-347

Query 6: I recommend checking the use of commas, as there are some missing commas, especially in the discussion.

We proceeded to a fully review of punctuation

Round 2

Reviewer 1 Report

The authors replied to my questions,